# Medfly Population Suppression through Augmentative Release of an Introduced Parasitoid in an Irrigated Multi-Fruit Orchard of Central–Western Argentina

**DOI:** 10.3390/insects14040387

**Published:** 2023-04-16

**Authors:** Lorena Suárez, María Josefina Buonocore Biancheri, Fernando Murúa, Mariano Ordano, Xingeng Wang, Jorge Cancino, Flavio Roberto Mello Garcia, Guillermo Sánchez, Sergio Beltrachini, Luis Ernesto Kulichevsky, Sergio Marcelo Ovruski

**Affiliations:** 1Dirección de Sanidad Vegetal, Animal y Alimentos de San Juan (DSVAA)-Gobierno de la Provincia de San Juan, Nazario Benavides 8000 Oeste, Rivadavia 5413, Argentina; 2CCT CONICET San Juan, Av. Libertador Gral. San Martín 1109, Capital 5400, Argentina; 3Planta Piloto de Procesos Industriales Microbiológicos y Biotecnología (PROIMI-CONICET), División Control Biológico, Avda. Belgrano y Pje. Caseros, San Miguel de Tucumán 4000, Argentina; 4Fundación Miguel Lillo, Miguel Lillo 251, San Miguel de Tucumán 4000, Argentina; 5Instituto de Ecología Regional, Universidad Nacional de Tucumán—Consejo Nacional de Investigaciones Científicas y Técnicas (CONICET), Las Cúpulas S/N, Horco Molle, Yerba Buena 4107, Argentina; 6USDA-ARS Beneficial Insects Introduction Research Unit, Newark, DE 19713, USA; 7Programa Moscamed-Moscafrut, SADER-SENASICA, Dirección General de Sanidad Vegetal, Camino a los Cacahotales s/n, Metapa de Dominguez 30860, México; 8Departamento de Ecologia, Zoologia e Genética, Instituto de Biologia, Universidade Federal de Pelotas, Pelotas 96000, Rio Grande do Sul, Brazil

**Keywords:** *Ceratitis capitata*, *Diachasmimorpha longicaudata*, biological control, eco-friendly control strategy, fruit crop, semi-arid environment, San Juan, Argentina

## Abstract

**Simple Summary:**

*Ceratitis capitata*, commonly known as the Mediterranean fruit fly or medfly, is one of the most damaging invasive insect pests for fruit production and trade worldwide. Augmentative biological control (ABC) using parasitoids, insects whose larvae develop as parasites that eventually kill their hosts, is an eco-friendly strategy for medfly suppression and is used in several countries. This strategy relies on periodic releases of large numbers of mass reared parasitoids in an area where naturally occurring parasitoids are too few to control the target pest. ABC has been incorporated for 10 years into medfly suppression strategies in the irrigated fruit production valleys of San Juan province, northwestern Argentina, by the San Juan Fruit Fly Control and Eradication Program. In this study, we report the results of mass releases of the Southeast Asia-native parasitoid *Diachasmimorpha longicaudata* in a fruit farm from early summer (December) to mid-autumn (May) over two successive years. The effect of the released parasitoids on the suppression of the medfly population was evaluated based on captures of adult flies in food-baited traps, and the resultant levels of parasitism of fly larvae were assessed based on adult flies in sentinel fruit traps. Our results showed a substantial decrease in the medfly population on the parasitoid release farm. Therefore, ABC can be an effective tool for medfly control in San Juan.

**Abstract:**

Biological control through the augmentative release of parasitoids is an important complementary tool that may be incorporated into other strategies for the eradication/eco-friendly control of pest fruit flies. However, not much information is available on the effectiveness of fruit fly parasitoids as biocontrol agents in semi-arid and temperate fruit-growing regions. Therefore, this study evaluated the effect of augmentative releases of the larval parasitoid *Diachasmimorpha longicaudata* (Ashmead) on *Ceratitis capitata* (Wiedemann) (medfly) populations over two fruit seasons (2013 and 2014) on a 10 ha irrigated fruit farm in San Juan province, central–western Argentina. The parasitoids were mass reared on irradiated medfly larvae of the Vienna-8 temperature-sensitive lethal genetic sexing strain. About 1692 (±108) parasitoids/ha were released per each of the 13 periods throughout each fruit season. Another similar farm was chosen as a control of non-parasitoid release. The numbers of captured adult flies in food-baited traps and of recovered fly puparia from sentinel fruits were considered the main variables to analyze the effect of parasitoid release on fly population suppression using a generalized least squares model. The results showed a significant decrease (*p* < 0.05) in the medfly population on the parasitoid release farm when compared to the Control farm, demonstrating the effectiveness of augmentative biological control using this exotic parasitoid. Thus, *D. longicaudata* could be used in combination with other medfly suppression strategies in the fruit production valleys of San Juan.

## 1. Introduction

*Ceratitis capitata* (Wiedemann), commonly known as the Mediterranean fruit fly or medfly, is one of the most damaging invasive insect pests for fruit production and trade worldwide [1,2]. Economic damage to the fruit is caused directly, both via pathogen transmissions through the oviposition of female flies and via larval feeding inside the fruit [3]. The medfly can also cause indirect economic losses due to quarantine restrictions imposed by importing countries [4]. This has led to the implementation of internationally recommended standard phytosanitary measures and treatments to avoid any ban on or rejection of fruit exports [5]. Endemic to Sub-Saharan Africa [6], *C. capitata* has spread worldwide, largely due to its extreme polyphagy [7], wide-ranging thermal tolerance [8], high reproductive capacity, mobility, and multivoltine reproduction [9], as well as its genetic evolution and adaption strategy during its establishment in a new region [10]. These traits have encouraged the pest’s fast invasion of new areas and infestation of novel hosts, which have brought about its successful establishment not only in wild environments, but mainly in highly diverse suburban and urban landscapes [11]. This invasive pest was first detected in 1905 in Argentina, infesting peaches in the province of Buenos Aires [12]. Since then, C. *capitata* has spread to all fruit-growing regions of the country, mainly due to the internal fruit trade [13]. It has successfully adapted to different climates, from the warm subtropical north, the dry temperate west, the humid temperate east, and the dry cold south in Argentina. Fifty-eight fruit species, including those from crop and non-crop plants, as well as introduced and indigenous plants, have been damaged by this exotic tephritid throughout the country [14,15]. This has severely affected Argentine fruit production, marketing, and export, and constrained the development of novel fruit crops and the expansion of fruit-growing areas [16].

Faced with the destructive impact of the medfly on the Argentine fruit industry, the National Agri-Food and Animal Health and Quality Service of Argentina (SENASA, Spanish acronym) established the National Fruit Fly Control and Eradication Program (PROCEM, Spanish acronym) at the beginning of the 1990s [13]. PROCEM’s current actions are focused on an area-wide integrated fruit fly management approach (AW-IFFM), suited to the ecological–geographical features of each fruit-growing region. Within this framework, in semi-arid temperate central–western fruit-producing regions, namely, the provinces of San Juan and Mendoza, control/eradication actions for medflies are based on the integrated use of the Sterile Insect Technique (SIT), chemical, cultural and biological control methods, quarantine protection systems, and a phytosanitary emergency schedule for pest outbreaks [13,17,18]. Both environmental conditions and landscape structure, with the Andean Mountain range bordering Chile on the west, influence medfly spread by restricting their populations to cultivated, suburban, and urban areas with host fruits under irrigation [19]. These areas are true oases within a semi-desert landscape matrix characterized by a combination of high arid mountains and plains covered by xerophytic vegetation. Therefore, the combination of the foregoing strategies has proven to be effective in achieving fruit fly pest-free or low-prevalence areas in this particular landscape framework [13,19]. This is particularly relevant because the productions of wines, raisins, grape (*Vitis vinifera* L.) must, fresh table grapes, olives (*Olea europaea* L.), olive oil, stone fruits (apricot (*Prunus armeniaca* L.), peaches (*Prunus persica* (L.) Stokes) and plums (*Prunus domestica* L.)), figs (*Ficus carica* L.), and quince (*Cydonia oblonga* Mill.) paste and jellies are powerful drivers of regional economic growth [19].

Augmentative biological control (ABC) using parasitoids was incorporated 10 years ago into the medfly management strategy of San Juan province by the San Juan Fruit Fly Control and Eradication Program (ProCEM—San Juan, Spanish acronym) [20]. This biological control strategy is based on periodic releases of mass reared parasitoids in a particular area with much larger numbers of wasps than naturally occurring ones, so that the area invaded by the target pest is inundated with the released natural enemy [21]. This is a suitable method for use when native natural enemies are absent or they are ineffective in exerting control over the pest, or when the ecological and/or geographic conditions of the area constrain the spread of both the pest and the released natural enemies [22,23,24]. This applies to all ecologically isolated and irrigated fruit-growing valleys in San Juan, where there are only common generalist predators such as ants, which usually attack medfly puparia, or spiders, which capture flying medfly adults, without having a significant effect on the pest population. In addition to this context, biological control as a well-used strategy is particularly desirable due to the following five key issues [25]: first, it is safe for the environment and health, sustainable as it slows down the development of pest resistance, inflicts no phytotoxic damage on the crop, produces higher yields and a healthier product, and reduces pesticide residues; second, it is a developed biological control industry that facilitates the large-scale and inexpensive mass production of biocontrol agents, with suitable quality control, and effective packaging, distribution, and release methods; third, biological control is effective in saving crops in agricultural production when pesticides are not available because of environmental or human health concerns; fourth, there are requirements, mainly from non-governmental organizations and consumers, to reduce pesticide residues below the currently permitted thresholds; and fifth, there has been a global increase in policies aimed at the reduction in synthetic pesticides and/or their replacement with more biologically sustainable methods of pest management. Furthermore, a highly significant advantage of biological control is its compatibility with environmental and health standards, which allows growers to link with integrated pest management (IPM) and organic certification schemes [26]. For this reason, the Southeast Asia-native parasitoid *Diachasmimorpha longicaudata* (Ashmead) (Hymenoptera: Braconidae) is currently mass reared at the San Juan Medfly and Parasitoid Mass Rearing Biofactory (known locally as BioPlanta San Juan) and is periodically released in cultivated suburban sites to achieve suppression or help with efforts to eradicate medfly populations [27]. This exotic braconid opiine is a solitary larval fruit fly endoparasitoid, frequently used as a tephritid biocontrol agent in augmentative release programs worldwide [23,28,29,30].

The establishment of *D. longicaudata* in the Americas has encouraged augmentative biological control development not only against the exotic *C. capitata* [30,31,32], but also against economically important native *Anastrepha* species, such as *Anastrepha suspensa* (Loew) [33,34], *A. ludens* (Loew), *A. obliqua* (McQuart), *A. serpentina* (Wiedemann), *A. striata* (Schiner) [23,35,36], and *A. fraterculus* (Wiedemann) [37]. This exotic braconid parasitoid was not found to attack non-target hosts or beneficial insect species in the American countries where it was released [30]. For instance, releases of *D. longicaudata* in disturbed wild areas of southern Mexico did not affect the richness of native parasitoid species, and the attack was focused on *Anastrepha* species [23]. In contrast to *D. longicaudata*, the cosmopolitan pupal parasitoid *Pachycrepoideus vindemiae* Rondani (Hymenoptera: Pteromalidae), which has also been used in field releases against tephritid species in the Americas, attacks the puparia of many cyclorrhaphous Diptera, and may also develop as a facultative hyperparasitoid of other primary tephritid fruit fly parasitoids (30). Previous studies have shown the promise of *D. longicaudata* mass reared at BioPlanta San Juan for the control of medflies in San Juan’s fruit-growing oases [20,38]. Given the isolated nature of these habitats, it is hypothesized that the augmentative release of *D. longicaudata* females in a particular area will likely reduce the medfly population confined to the area when compared to the non-release area. Therefore, this study aimed to assess the effect of *D. longicaudata* releases on the suppression of medfly populations based on adult flies captured in food-baited traps and fly puparia recovered from sentinel fruits during the release periods in two different study seasons and weather conditions. Such an assessment has become relevant because different field tests with *D. longicaudata* and other parasitoid species for the suppression of medfly populations have mainly been carried out in tropical or subtropical environments [30,31,32,39,40,41,42,43].

The study herein reported is the result of an ongoing effort initiated in 2008 by the San Juan government through ProCEM—San Juan. The goal was to achieve the development and application of augmentative biological control in a framework of environmentally friendly strategies to suppress medfly populations in the irrigated fruit-growing valleys of San Juan. Therefore, the findings from this research are discussed with respect to the use of the exotic *D. longicaudata* against medflies in the fruit-growing central–western region of Argentina featuring a semi-arid climate, in which pest populations are focused in irrigated urban and suburban areas with a high abundance of exotic host plants. Both climatic conditions and landscape structure encourage pest proliferation in such regions.

## 2. Materials and Methods

### 2.1. Source of Insects and Rearing Procedures

The cohorts of *D. longicaudata* used in this study were taken from the Parasitoid Rearing Laboratory of the BioPlanta San Juan mass rearing facility of the San Juan Biotechnology Center of the Dirección de Sanidad Vegetal, Animal y Alimentos (DSVAA, Spanish acronym) of the government of the San Juan province. This governmental institution dedicated to agricultural crop and livestock health is located in the Rivadavia District, San Juan Province, in central–western Argentina. Adult *D. longicaudata* were reared on 90 Gy-irradiated third instars of the Vienna-8 temperature-sensitive lethal (tsl) genetic sexing strain of *C. capitata*. This lineage of the *D. longicaudata* population is hereafter referred to as *Dl*_TSL-*Cc*_. The parasitoid colony was kept in 60 × 60 × 30 cm rectangular iron-framed mesh-covered cages at 24 ± 1 °C, at 65 ± 5% RH and under a 12 h photoperiod. Adult parasitoids were fed honey and water *ad libitum* every other day. Cohorts from the *D. longicaudata* colony at their 52nd–64th generations under mass rearing conditions were used in open-field experimental releases. The female and male parasitoids released were 1–2 and 2–3 days old, respectively. Larvae of the Vienna-8 *C. capitata* strain were reared in the Medfly Rearing Laboratory at BioPlanta San Juan, according to the medfly tsl strain rearing protocol published by Cáceres [44].

### 2.2. Experimental Locations and Study Area Description

Two 10 ha multi-fruit farms, with a predominance of grapes for fresh consumption and for basic wines, musts, and raisin production, were selected as study sites. Both farms are located in a suburban area of the Rivadavia District, neighboring BioPlanta San Juan. Backyard fruit and various local orchards were the only sources of medflies in the study area. Farm #1, hereinafter referred to as Treatment Farm or “TF”, was located at 31°31′46″ S and 68°38′29″ W, and at 2.6 km in an east–south direction from BioPlanta San Juan, while farm #2, henceforth named Control Farm or “CF”, was situated at 31°31′39.1″ S and 68°38′22.2″ W, and at 2.5 km west–south from BioPlanta San Juan (Appendix A). The farms were 5 km apart and located at an altitude of 723 m above sea level. The TF was characterized by fruit species such as plum (80 trees), fig (12), quince (6), olive (9), sweet orange (*Citrus sinensis* (L.) Osbeck) (1), loquat (*Eriobotrya japonica* (Thunb.) Lindl.) (1), rose (*Rosa canina* (Bastard) Rapin) (5), commercial grape crops of the “cherry” and “Muscat of Alexandria” varieties (~7 ha), and half a ha of casuarina (*Casuarina equisetifolia* L.) tree crop (Appendix A). Grapes, figs, quinces, and olives were fruiting at the time of parasitoid release. The TF was surrounded by windbreak forest curtains composed of casuarina and common poplar (*Populus alba* L.) trees. The farm was bounded to the east by uncultivated land with xeric shrubs, to the south by both xerophytic vegetation and urban areas, to the west by grape crops and houses with spacious back yards, and to the north by a highly built-up area (Appendix A). The CF was characterized by plum (3 trees), peach (3), apricot (1), fig (10), quince (11), olive (30), sweet orange (1), grapefruit (*Citrus paradisi* Macfadyn) (1), tangerine (*Citrus reticulata* Blanco) (1), lemon (*Citrus limon* (L.) Burm.f.) (2), rose (7), and “cherry” and “Muscat of Alexandria” grape crops (~8 ha) (Appendix A). Similarly to the TF, the fruit species at the fruiting stage during the study were grapes, figs, quinces, and olives. This farm was partially surrounded by common poplar trees. It was bounded to the east by mainly uncultivated but cleared land, to the south by olive and grape crops, to the west by largely uncultivated land with xeric shrubs and small farms with grape crops, and to the north by houses with large back yards covered with shade trees and some fruit species (Appendix A). The fruits of these farms, except grapes, are generally used for the production of jams and jellies and quince paste, as well as for local fresh consumption. The fruits from both farms were not chemically treated with any pesticide during the parasitoid releases. Only herbicides were used for weed control in the commercial grape crops.

This area was not undergoing control against medflies by ProCEM San Juan during the periods in which the research was carried out. Records on the number of caught medfly adults in the suburban areas of both TF and CF one year before the study revealed high fluctuations in the fly population between mid-spring and late autumn. Thus, the number of caught flies per trap and per day (FTD index) ranged from 0.04 ± 0.04 to 0.89 ± 0.64 and from 0.09 ± 0.09 to 0.99 ± 0.30 in both the TF and CF, respectively.

The climate in the study area is temperate–dry with very marked thermal amplitudes during both the day and the year, and experiences scarce rainfall throughout the year, mostly focused in the summer (December–March). The average annual temperature and rainfall are ~17 ºC and ~100 mm, respectively. In the west, the Andes mountain range acts as a barrier against the humid and cold winds from the Pacific.

### 2.3. Parasitoid Release Procedure

Parasitoid releases were carried out to assess the effectiveness of *D. longicaudata* in controlling the medfly population in the TF. For this purpose, about 20 cm^3^ of parasitized *C. capitata* puparia, the equivalent of 1,200 puparia, were kept in a sulfite paper bag (17 × 49 cm, width × height). A strip of laboratory blotting paper (3 × 30 cm, width × height) filled with icing sugar as the food was placed inside the bag, which was then closed at the top with six staples. Thirty bags per release date were prepared as described above and stored in a room at 25 ± 1 °C and 70 ± 5% RH for 3 days. Both male and female parasitoids emerged during this time. The percentage and sex ratio of the wasps that emerged from each bagged puparium were recorded on the release day. The percentage of *D. longicaudata* that emerged under the packaging conditions ranged from 44 to 50%, with a sex ratio of 0.8–1.1 females per 1 male. Thus, each release bag held between 528 and 600 parasitoids. The bags were taken in an air-conditioned vehicle to the TF for each release. Adult parasitoids were released on 13 different dates between 26 December, 2012 (early summer) and 23 May, 2013 (mid-autumn) and between 26 December, 2013 and 23 May, 2014. The release interval was 12 to 13 days. Parasitoids were released from the ground in 10 release sectors spaced 40 m apart and located along three 400 m long line transects in a south–north direction (Appendix A). One transect was located at the center of the TF, one near the western edge, and another one close to the eastern edge of the farm. Transects were separated from each other by ~80 m; they were 20 m apart from both the western and eastern edges and 50 m from both the southern and northern edges of the farm, covering 6.4 ha (Appendix A). On the release day, one bag per release sector was opened along each transect. Soaked blotting tissues and others with icing sugar were put on tree branches as a food resource in each release sector. Between ~16,000 and ~18,000 parasitoids (~1692 ± 108 parasitoids/ha, including ~803.7 ± 126.9 females/ha, mean ± SE) were released on the TF per release time. During the parasitoid release period on the TF, there were some flowering garden plants, such as *Camellia* L. (Theaceae), which blooms from early summer (December) to late summer (March), and rose gardens, which bloom from mid-spring (October) to late summer. These flowers probably provided a food source for the released parasitoid population. 

### 2.4. Fly and Parasitoid Monitoring

The *C. capitata* populations on both the TF and CF were monitored using both McPhail traps (Susbin^®^, Mendoza, Argentina) and sentinel fruit “traps” placed in artificial devices made of disposable plastic bottles. This was carried out to compare medfly population fluctuations between the two study sites over time. Fruit traps were also used to record levels of *D. longicaudata* emergence and host pupal mortality. This also provided an assessment of the impact of the parasitoid releases on the pest population.

The McPhail trap had a yellow base and a transparent lid made of high-strength UV-filtered plastic. The traps were baited with Susbin PBX^®^ torula yeast pellets plus water as a food attractant for both males and females. Four pellets per 200 cm^3^ of water were used. Three traps were placed along a 500 m long line transect in a south–north direction and located in the central part of the TF (Appendix A). The traps were separated from one another by 165 m; they were 100 m away from both the western and eastern edges and 85 m away from both the southern and northern edges of the farm (Appendix A). Three other traps were positioned along a 670 m long line transect in a south–north orientation and located at the center of the CF (Appendix A). On this farm, each trap was spaced 235 m apart from the next one, and were 75 m away from both the western and eastern edges and 100 m away from both the southern and northern edges of the farm (Appendix A). The traps were placed either in the middle portion of the fruit tree canopy or inside the vineyard at a 1.5–2 m height. The traps remained in place for 24 weeks between 26 December 2012 and 11 June 2013 and between 26 December 2013 and 11 June 2014. The traps were checked every 7–8 days. Caught flies were removed and the trap was rebaited with fresh attractant during each check. All around the TF and CF, 14 liquid McPhail traps baited with Susbin PBX^®^ torula yeast pellets were used to isolate the medfly population on each farm, and to prevent, as far as feasible, the access of adult flies from neighboring areas. Traps were placed parallel to the farm edges, ~30 m away from them, and spaced 150 m apart (Appendix A).

The artificial devices used to hold sentinel fruits and expose them to natural infestation by wild *C. capitata* females were similar to those used by Sanchez et al. [20] (but slightly modified) (Appendix A). This oviposition device, hereinafter named “OD”, consisted of a 200 mL longitudinal disposable plastic bottle with wheat bran inside to provide a pupation substrate. The central part of the bottle had a 45 × 15 cm (length × width) rectangular hole. At each end of the hole was a galvanized wire frame (14-caliber and 2 mm diameter) in an inverted U-shape piercing the bottle. Each structure extended upward with a 12 cm long wire, whose top was bent into a hook shape to cling to the plant branch and hold the device in place. One peach (*P. persica*, Elegant Lady variety) was placed on a 10 × 15 × 0.6 cm (length × width × height) heavy-duty plastic ventilation screen attached to both sides of the container with galvanized wire (Appendix A). Peaches, like figs, are a primary host that encourage medfly proliferation throughout most of the fruit-growing Argentinean regions [12]. Only one peach per OD was placed. The fruit was fastened to the plastic screen using a 1 cm wide Parafilm “M”^®^ strip (Pechiney Plastic Packaging, Chicago, USA), which was wrapped around the sides and the bottom of the bottle. The peaches used in the ODs were consistent in size, weight, ripeness grade, and physical features. Round, ripe, yellowish-greenish peaches with slight reddish tones, with a weight of 141.2 ± 0.3 g and an equatorial diameter of 6.4 ± 0.8 cm (mean ± SE), were used. All peaches were bought at a local organic greengrocer store, and before using, were washed first with a sodium benzoate plus methyl p-hydroxybenzoate solution, and then, twice with water only.

Each OD was hung 1.5 m from the ground level, either from an inner branch of a fruit tree (fig, quince or plum) or inside the vineyard 2 m from the alleyway. Ten ODs were distributed on both the TF and CF in three line transects (Appendix A) 5–6 days before each parasitoid release date. Overall, the sentinel fruit remained exposed for 12 to 13 days on the farms, i.e., the fruit was exposed for a full period between two consecutive parasitoid releases, and then, replaced with a new fruit. The two transects were 140 m long. The first one had a south–north orientation, while the second one had a west–east orientation. In each of these rows, three ODs were placed 70 m apart from one another (Appendix A). The third transect was 210 long with a south–north orientation and housed four ODs spaced 70 m apart. The first transect was located in the northern sector of the farms, approximately 100–120 m and 75–80 m from the western and eastern edges of the TF and CF, respectively (Appendix A). The second transect was established in a more central sector of both farms, 250 m and 330 m from the northern and southern edges of the TF and CF, respectively (Appendix A). The third transect was located in the southern sector of the two farms, 30–35 m from the western edge. The devices covered around 58 and 61% of the total area of the TF and CF, respectively (Appendix A).

### 2.5. Traps and OD Laboratory Processing

Trapped flies were identified in the laboratory, and the FTD index was calculated based on weekly captures of flies and used as a response variable in the study. The pupation substrate of each OD was sieved to recover puparia originating from larvae that fell from the peach. These puparia and their respective peaches from the devices were placed in 500 mL plastic containers with sterilized wheat bran at the bottom. The container lid was covered with voile for ventilation. All containers were placed in a room at 26 ± 1 °C, 75 ± 5% RH, with a light/dark ratio of 10:14 (L:D) h. The fruit was removed from each container after 5 days, and dissected to recover the larvae and/or puparia, which were returned to the container until adult flies or parasitoids emerged. Thus, the numbers of recovered medfly puparia, emerged medfly adults, emerged parasitoid adults, and non-emerged medfly puparia per OD were recorded. Other response variables for data analysis were calculated using these data, such as the percentages of medfly emergence, parasitism, and medfly pupal mortality. Medfly emergence was determined as the number of emerged fly adults divided by the total number of puparia recovered from the OD × 100. Parasitism was calculated as the number of emerged parasitoid adults divided by the total number of puparia recovered from the OD × 100. Medfly pupal mortality was determined as the number of non-fly-producing puparia divided by the total number of puparia recovered from the OD × 100.

### 2.6. Weather Conditions during the Study

The mean temperature, relative humidity, and cumulative rainfall variations recorded during both experimental periods are shown in Figure 1. The meteorological data were recorded by digital weather stations (WS-80, LUFT^®^, Shenzen, China) located on each farm.

### 2.7. Statistical Analyses

#### 2.7.1. Database Management and Statistical Descriptions

Given the longitudinal design of comparing the two different scenarios, with and without parasitoid release, and the use of different methods to describe the variables of interest, the period between two consecutive releases within each study season was established as the data management unit. Two sources of data were used for data analysis. In one database, all the available response variables for each period, using the average, were combined. In this way, a database that describes FTD, those variables related to *C. capitata* and *D. longicaudata*, and weather variables, with the average value for each release period, was set up. This base of the averages made it possible to describe the central tendency and dispersion and to explore correlations between variables. For the correlations between variables, Pearson’s correlations were used on the averages of the variables by period. Then, the medfly emergence percentage was modeled as a function of the release treatment and temperature using a generalized least squares (GLS) model. Medfly emergence was chosen because it summarizes the effects of previous selective periods of the *C. capitata* life cycle, with which it is correlated (see results). Air temperature was chosen because it is a recognized predictor of insect development dynamics [45]. In the other database, all the individual data from the ODs were used, which made it possible to evaluate the performance of both *C. capitata* and *D. longicaudata* while considering the variation within each condition and each period. Generalized least squares models were applied to evaluate the effects of *D. longicaudata* releases on wild medfly populations on the TF during the two study seasons (2013 and 2014).

#### 2.7.2. GLS Modeling

The field sampling design throughout the 13 periods precluded the error independence due to the temporal structure of the design. In this way, the covariance among sampling periods is expected to differ from zero. To explicitly consider the temporal error structure in the model, GLS models were used [46,47]. This GLS model allows for the inclusion of a correlation function associated with a given error structure [48]. The database of averages was first used to model the proportion of medfly emergence as a function of the air temperature. Thus, the percentage of medfly emergence was the response variable. The treatment (release farm and non-release farm) and the study season were fixed factors in the model. In addition, the interaction between treatment and season, with temperature as a continuous predictor, was included. The nesting structure was the period nested in each season. To consider the variation in the medfly performance within each release period, another model using the database, with data from individual ODs, was built. In this model, medfly emergence was the response variable, and the condition (release farm or non-release farm), study season, and the release period (1–13 periods) were fixed factors. The interaction between release treatment and release period was also included, because the efficiency of treatment might be affected by the period within each season. Interactions with the season factor were not included in the model because they were non-significant. The nesting structure was the OD in nested in each period. The selection of medfly emergence as the main response variable simplifies the analyses given the positive and significant correlation with other medfly variables (see Results). In both modeling approaches, an error structure associated with an ARMA structure was chosen, which considers both the autoregressive process given by the periods (*p* = 1) and the moving average process given by the seasonal effects (q = 1) [49]. The models were fitted with restricted maximum likelihood. The “nlme” package [50], among others, was used in R [51]. The R Scripts are provided as supplementary files (“medfly.Rmd”).

## 3. Results

### 3.1. Statistical Description and Correlations among Tested Variables

Overall, higher numbers of both medfly puparia and adults were recorded under non-release conditions in both study seasons (Table 1). Adult *D. longicaudata* were recovered from medfly puparia in both release seasons on the TF farm, and the parasitism rate was nearly 1.5-fold that of the second season (Table 1).

The temporal variation in the number of emerged medfly and parasitoid adults and the number of recovered medfly puparia during the release periods in the two seasons are shown in Figure 2A. A higher number of emerged medfly adults from the fifth release period of the first season (2013) onwards was noted on the CF relative to the TF. This difference in the number of emerged medfly adults between TF and CF was more pronounced from the first release period in the second season (2014). The number of emerged parasitoid adults was constant throughout release periods in both study seasons, except for a drop in the last three release dates. The number of medfly puparia recovered on the CF was mostly higher than that puparia recovered on the TF from the sixth release period onwards in the first season. In contrast, in the second season, this difference between the TF and CF regarding the number of recovered medfly puparia was more evident from the first release period onwards. When considering the data on medfly adults that emerged per release period as proportions, based on the total number of puparia recovered, the difference in the temporal variation in medfly emergence is even more perceptible between the TF and CF in the two study seasons (Figure 2B). Parasitism was more variable throughout release periods and had lower values in the first study season than in the second season (13.5–26.8% vs. 18.3–35.5%) (Figure 2B). In both seasons there was an increase in parasitism levels at the beginning of the parasitoid release periods and a slight decrease from the tenth release period onwards. The pattern of variation in medfly pupal mortality (total number of medfly puparia from which no flies emerged) on the TF was similar in the two study seasons, with a gradual increase from the beginning to the last release period (Figure 2B). There was also an increase in medfly pupal mortality on the CF from the beginning to the end of the release period, although this pattern was even less pronounced than that found on the TF (Figure 2B). Medfly pupal mortality on the TF was close to 2-fold that recorded on the CF.

Medfly emergence positively correlated with the majority of the variables tested, such as the number of recovered medfly puparia and emerged adults, the FTD index, and air temperature (Table 2). The correlations between parasitism and the number of released female parasitoids and weather conditions were not significant (Table 2).

### 3.2. Medfly Performance and Temperature

The GLS model applied to the database with averages by release period shows that medfly emergence is slightly affected by the covariate “air temperature” (coefficient = 0.012 ± 0.001 (SE); *F*_1, 47_ = 46.310; *p* < 0.0001). This effect is probably driven by the variation between release periods within each study season. In addition, when comparing the variation in air temperature between seasons, medfly emergence was slightly higher in the 2014 season (*F*_1, 47_ = 10.690; *p* = 0.002). The parasitoid release condition also had a negative effect on medfly emergence, as there was a significant drop in the proportion of emerged medfly adults on the TF (coefficient = −0.259 ± 0.044; *F*_1, 47_ = 84.510; *p* < 0.0001), even when accounting for variation in air temperature. The interaction between treatment and season was not significant (coefficient = −0.081 ± 0.065; *F*_1, 47_ = 1.565; *p* = 0.2171) (Figure 3).

### 3.3. Medfly Performance and Temporal Variation

The GLS model applied to the database, with full data from individual ODs within each release period, shows that medfly emergence was affected by both the release treatment (*F*_1, 12_ = 831.191; *p* < 0.0001) and release period (*F*_1, 12_ = 14.445; *p* < 0.0001) (Figure 4). Medfly emergence was not affected by the study season (*F*_1, 12_ = 0.055; *p* = 0.8143). The previous analysis indicated that by controlling for the covariate air temperature, it was possible to detect air temperature effects on medfly emergence between seasons. In this case, the effects of the period reduce the variation in medfly emergence; this could depict intra-annual changes in weather conditions driven by the variation between summer and autumn, regardless of the variation between years or experimental seasons. There were also significant effects of the interaction between release treatment and period (*F*_1, 12_ = 5.131; *p* < 0.0001).

## 4. Discussion

The success of any AW-IFFM program depends heavily on the implementation of coordinated management strategies aimed at eradicating/controlling the target pest while minimizing environmental impact, maximizing human and non-target organism care, and ensuring long-term sustainability. Biological control through augmentative parasitoid releases can be combined with other eco-friendly pest suppression techniques. Therefore, augmentative biological control has become an important complementary tool in those programs addressing different fruit fly eradication/control strategies with a broad, modern, and integrated approach. However, the ability of fruit fly parasitoids to act as effective biocontrol agents under open-field conditions requires further assessment. This knowledge is strategic and crucial for developing and implementing augmentative biological control against pest fruit flies. From this perspective, the current study reports the impact of *Dl*_TSL-*Cc*_ lineage augmentative releases on medfly populations during two consecutive study seasons in a fruit-producing irrigated farm in San Juan Province, central–western Argentina. Two major findings stand out in this study. Firstly, the decrease in the medfly population at the *D. longicaudata* release site provides evidence of the effectiveness of augmentative biological control using parasitoids on farms with little or no conventional chemical treatment. Secondly, parasitoids of the *Dl*_TSL-*Cc*_ lineage showed satisfactory performance once released under environmental conditions, prevailing in fruit orchards, as they successfully found and parasitized medfly larvae.

The above two findings are supported by three strongly correlated outcomes recorded from the two parasitoid release seasons on the TF, and compared with those from the CF. These results are: (1) the substantial decline in the FTD index; (2) the considerable decrease in the number of medfly puparia recovered from the fruit located in the ODs; and (3) the significant reduction in medfly adult emergence. The decreased medfly adult prevalence level on the TF, which was most pronounced in the second parasitoid release season (ca. 2 times lower than on the CT), may imply that there were fewer females ovipositing on fruits on the farm; this most likely induced a lower number of puparia recovered from fruit traps, and therefore, fewer emerged medfly adults. In this regard, the results from the two study seasons on the TF clearly show that the effect of periodic inundation releases of *D. longicaudata* may reduce adult pest emergence by nearly three times that recorded on the CF. These findings are in agreement with those reported in previous augmentative releases of the *Dl*_TSL-*Cc*_ lineage on a medfly-infested fig crop in San Juan [20]. The aforementioned releases caused a 1.5 and a 1.8-fold reduction in medfly emergence in the tested fig plots compared to the controls [20]. Interestingly, *D. longicaudata* mass releases were also significantly effective in reducing *C. capitata* populations within olive crops in Tacna, a region of Peru that borders Chile and has temperate–desert environmental conditions [30], similar to those of San Juan. All this information on the effectiveness of *D. longicaudata* in areas with semi-desert to desert climates is consistent with previously published data on the successful performance of this braconid parasitoid in reducing coffee-infesting medfly populations on the Mexico–Guatemala border [31,32]. Mass releases of *D. longicaudata* proved to be efficient not only against *C. capitata*, but also against various *Anastrepha* pest species in tropical and subtropical areas. Evidence of this is provided by the substantial population reduction of *A. suspensa* in Florida, USA [33,34], *A. ludens* and *A. obliqua* in Chiapas, México [23,35,36], and *A. fraterculus* in São Paulo, Brazil [37] at *D. longicaudata* mass release sites. Previously published information and the information reported in this study provide strong evidence of the benefit of using *D. longicaudata* strategically under an augmentative biological control approach. This is further strengthened by the high adaptability of *D. longicaudata* to regions of the world with different climates and host fruit species into which it has been introduced and released [30,38]. All of this, combined with the ability of *D. longicaudata* to successful develop in medfly larvae infesting a wide variety of fruit species under San Juan field conditions [20,38], and current study data, supports the large-scale use of this exotic parasitoid against *C. capitata* in the San Juan fruit-growing valleys and in regions with similar conditions.

The positive correlation between air temperature and pest emergence is an effect mostly driven by the decrease in medfly emergence from the tenth release period when the coldest and driest conditions began in the study region. This gradual decrease in the pest population between release periods 10 and 13 on both the TF and CF in the two seasons is consistent with the FTD data recorded for autumn–winter by the PROCEM—San Juan trapping network for medfly monitoring within the suburban and rural area. However, the reduction in medfly emergence on the TF during the last release periods is markedly more pronounced than that recorded on the CF, due undoubtedly to the effect of *D. longicaudata* releases. Surprisingly, there was no significant influence of weather factors, such as temperature and/or relative humidity, on the parasitism of *C. capitata* by *D. longicaudata*. This information contrasts with a previous study conducted in San Juan, which found that the performance of released *D. longicaudata* females increased at higher temperature and relative humidity values [20]. All this information highlights the need for further study on the bioclimatic requirements of *D. longicaudata* in San Juan. In this regard, field-cage studies on diapause, longevity, and fecundity in this exotic braconid species are currently being carried out in San Juan. Nevertheless, augmentative parasitoid releases, particularly during the period of medfly population growth (late spring–early summer, i.e., between November and December) [52], could favor pest reduction, mainly in urban and suburban areas where chemical control is restricted. Based on medfly adult emergence levels, parasitoid releases may be performed up to the 11th release period (April 28, early autumn) because after this date, it makes no economic sense to release parasitoids. This is because the parasitoid releases did not have a substantial impact on the pest population in relation to unfavorable climatic conditions, which affected the performance of both the natural enemy and the pest.

In conclusion, the current study supports earlier results on *D. longicaudata* releases in San Juan and proves the success of augmentative releases of this exotic parasitoid species in terms of reducing medfly populations in irrigated orchards under a highly dry environment with wide thermal variation, as characterized by a fruit-growing central–western region of Argentina. This study demonstrates that *D. longicaudata* is a helpful tool for medfly population suppression in fruit-producing areas of San Juan, a founding purpose of ProCEM—San Juan [25]. Whereas *D. longicaudata* is a generalist parasitoid of at least 34 fruit fly species [30,53,54], the medfly is the only pest tephritid species in San Juan, and its populations are focused in both urban and rural artificial irrigation oases in a desert landscape. Such a scenario justifies the release of this generalist exotic parasitoid in San Juan as a control agent of the invasive pest *C. capitata*. Thus, the use of augmentative parasitoid releases associated with SIT and/or mass trapping, both of which are strategies used by PROCEM—San Juan within an AW-IFFM approach [19,20], should be encouraged.

## Figures and Tables

**Figure 1 insects-14-00387-f001:**
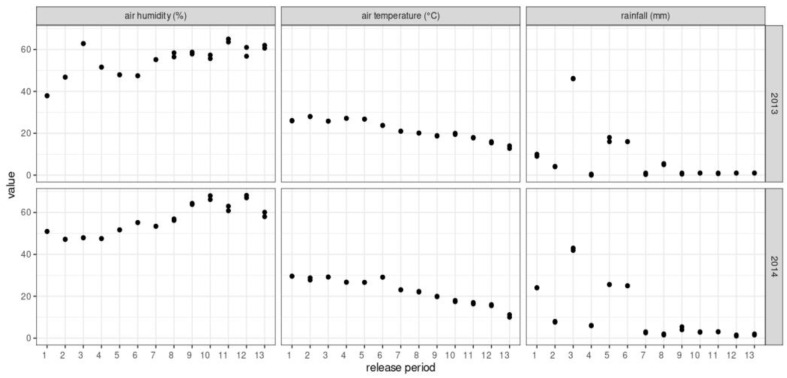
Mean (±SD) values of temperature (Temp.) and relative humidity (RH), and cumulative rainfall (CR) recorded every 12 days between December and June of both study seasons (December 2012–June 2013 and December 2013–June 2014) on two experimental irrigated fruit farms in Rivadavia District, San Juan Province, central–western Argentina.

**Figure 2 insects-14-00387-f002:**
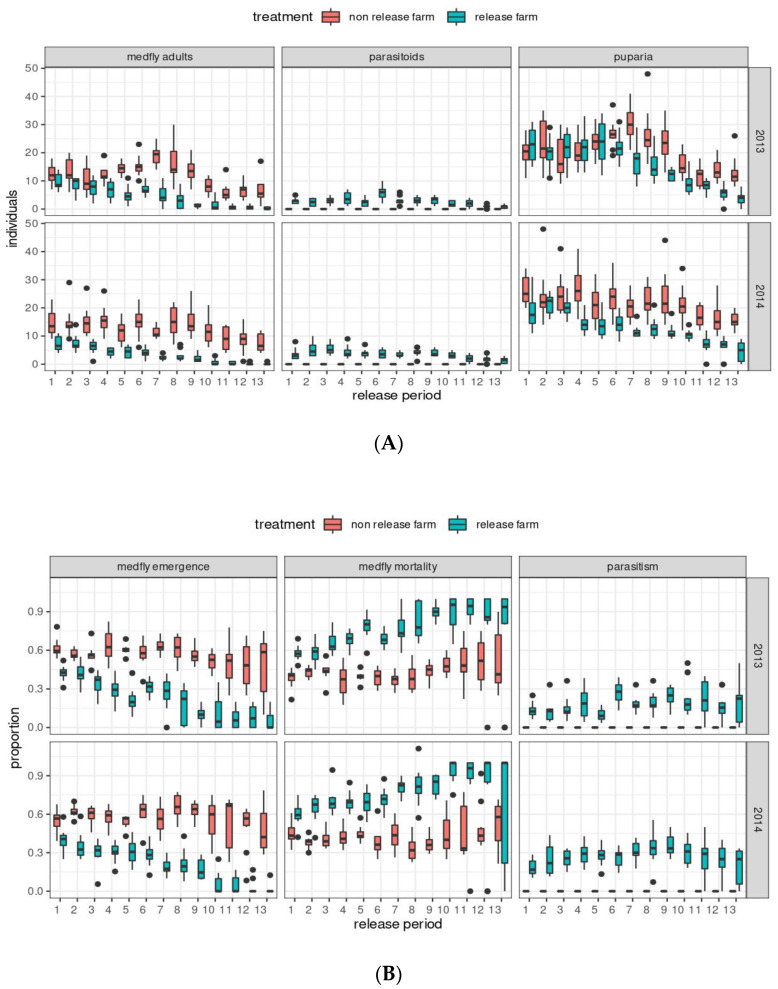
Temporal variations in the numbers of emerged medfly (*Ceratitis capitata*) and parasitoid (*Diachasmimorpha longicaudata*) adults, and recovered medfly puparia (**A**), and proportions of medfly emergence, parasitism, and medfly mortality (**B**) by release period (between 26 December 2012 and 23 May 2013, and between 26 December 2013 and 23 May 2014) and farm condition (non-parasitoid release farm (CF) and parasitoid (*D. longicaudata*) release farm (TF)). The boxes signify the interquartile range and include the median; the whiskers characterize data 1.5 times the interquartile range, and the black points are outliers.

**Figure 3 insects-14-00387-f003:**
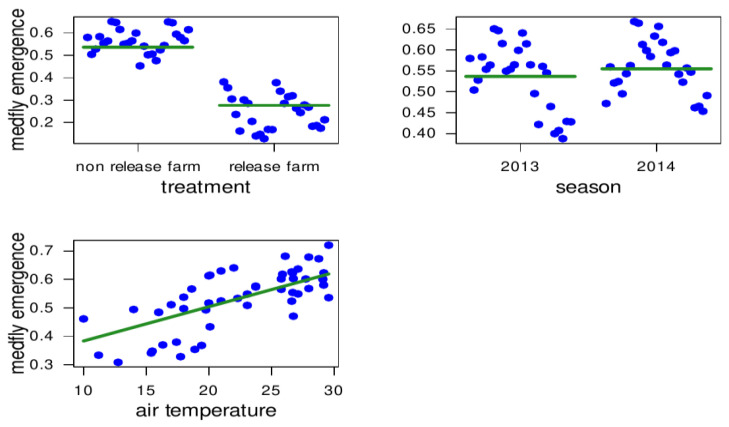
Direct influences of treatment (non-parasitoid release farm (CF) and parasitoid (*Diachasmimorpha longicaudata*) release farm (TF)), study season (years 2013 and 2014), and air temperature on medfly (*Ceratitis capitata*) emergence (proportion of emerged medfly adults) on two experimental fruit farms in San Juan, central–western Argentina. The fitted green central line depicts partial regression from the GLS model applied to averaged values by period, while the blue points are the observed ones. The partial residuals show that the magnitude of medfly emergence on the parasitoid release farm is almost half that of medfly emergence in the non-parasitoid release farm. The distribution patterns of the errors within each group denote the time-structured variation within each season, that is, variation due to differences between groups within a season. The temperature has a positive effect on medfly emergence.

**Figure 4 insects-14-00387-f004:**
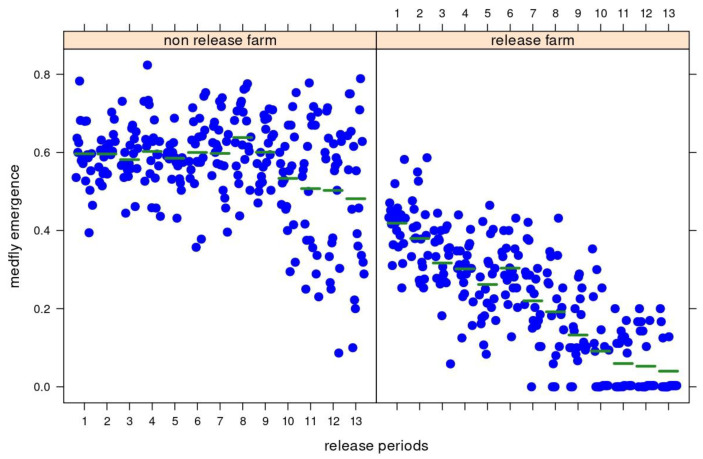
Direct influences of treatment (non-parasitoid release farm (CF) and parasitoid (*Diachasmimorpha longicaudata*) release farm (TF)) and release period (1 to 13 from summer to autumn) on medfly emergence (proportion of emerged medfly adults) on two experimental fruit farms in San Juan, central–western Argentina. The fitted green central line depicts the partial regression from the GLS model applied to individual data by period, while the blue points are the observed ones. The partial residuals show that the magnitude of medfly emergence decreases throughout periods within seasons, and the decreasing trend in medfly emergence is higher on the parasitoid release farm.

**Table 1 insects-14-00387-t001:** Mean (±SD) values of response variables tested under parasitoid release and non-release conditions. Mean and SD were estimated for the averaged database by release period.

Variables	Farm Conditions/Study Seasons			
	Non-Parasitoid Release		Parasitoid Release	
	2013	2014	2013	2014
No. of recovered medfly puparia	20.7 ± 5.9	21.9 ± 3.8	15.7 ± 7.2	12.8 ± 5.2
No. of emerged medfly adults	12.0 ± 4.2	12.7 ± 2.9	4.4 ± 3.4	3.2 ± 2.6
No. of emerged parasitoid adults	0	0	2.7 ± 1.3	3.5 ± 1.2
No. of non-emerged medfly puparia	8.7 ± 1.8	9.2 ± 1.3	8.6 ± 3.4	6.0 ± 1.6
Medfly emergence (%)	56.7 ± 5.2	57.2 ± 5.0	22.0 ± 12.9	20.4 ± 12.7
Parasitism (%)	0	0	18.3 ± 4.7	27.1 ± 4.7
Medfly pupal mortality (%)	43.1 ± 12.0	41.9 ± 11.4	76.2 ± 19.4	75.2 ± 22.0
FTD index	1.0 ± 0.2	1.1 ± 0.2	0.7 ± 0.3	0.6 ± 0.3
No. of released female parasitoids	0	0	8861.5 ± 312.1	8826.9 ± 251.3
Air temperature (°C)	21.9 ± 4.6	22.7 ± 6.2	21.8 ± 4.9	22.7 ± 6.2
Air relative humidity (%)	54.6 ± 7.8	56.2 ± 7.4	54.1 ± 7.4	56.1 ± 7.2
Accumulative rainfall (mm)	7.9 ± 12.3	11.5 ± 12.8	8.0 ± 12.5	11.4 ± 12.6

**Table 2 insects-14-00387-t002:** Summary of Pearson’s correlations between variables based on the averaged database by release period. The medfly (*Ceratitis capitata*) and parasitoid (*Diachasmimorpha longicaudata*) variables correspond to the correlations under non-parasitoid release and parasitoid release conditions, respectively.

First Response Variable	Second Response Variables	Person’s Correlation Results				
		*r*	Lower Confidence Limit	Upper Confidence Limit	*t*	*p*
Medfly emergence	No. of recovered medfly puparia	0.740	0.489	0.874	5.337	<0.001
	No. of emerged medfly adults	0.850	0.695	0.932	8.002	<0.001
	No. of non-emerged medfly puparia	0.360	−0.026	0.659	1.919	=0.067
	FTD	0.620	0.307	0.813	3.880	<0.001
	Air temperature	0.620	0.301	0.810	3.836	<0.001
	Air relative humidity	−0.460	−0.721	−0.091	−2.553	=0.017
	Cumulative rainfall	0.170	−0.231	0.524	0.855	=0.410
Parasitism	No. of released female parasitoids	0.210	−0.195	0.551	1.042	=0.308
	Air temperature	−0.089	−0.460	0.309	−0.436	=0.667
	Air relative humidity	0.290	−0.106	0.611	1.505	=0.145
	Cumulative rainfall	−0.110	−0.481	0.285	−0.566	=0.576

## Data Availability

The data presented in this study are available in the Appendix A (S8–S10).

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
