# Peer review of "Medfly Population Suppression through Augmentative Release of an Introduced Parasitoid in an Irrigated Multi-Fruit Orchard of Central–Western Argentina"

_insects, 2023, doi:10.3390/insects14040387_

Round 1

Reviewer 1 Report

Intorduction: Authors should give more extensive report on the other authors results of ABC experience with D. longicaudata: supression efficacy and host plants suitability.

Lines:

61, instead tephritid dipteran use taxonomic titles

94-98, add references

101-102 - add latin names of fruit species

121 - only reference 23 report use of D. longicaudata, references 26-28 not, please add correct reference

Materials and methods: Listing of the names and number of host plants per treated plots are confusing with local names, common names and latin names, please simplifie 

2.1. - age of parasitiods releases,  

190 - what means little conventional chmical treatments ?explain

198-201, state for which period is the FTD value reported, is it average for whole spring-autumn, since FTD is probably low in spring and high in autumn

add the size of experimental plot in scheme 2 and 3

2.3 section, explain better the release rate per release date/ per area size, not per bag. What is release rate females per ha ? What is the interval of the release, is it 2 weeks ?

Add the size of area in Sceheme 4

Results:

Author Response

Dear Reviewer 1

Suggested corrections and changes by you have improved the quality of the manuscript. We thank  for their time in reviewing our manuscript. Below you will find a point-by-point response to the requests made. Best regards, Sergio M. Ovruski (Corresponding author)

Reviewer #1.

  1. Introduction: Authors should give more extensive report on the other authors results of ABC experience with D. longicaudata: supression efficacy and host plants suitability.

Response: We have included a new paragraph in the Introduction in which we make references to what was requested by the reviewer (attached below). In addition, it is important to clarify that in the Discussion section we provide details of the effects of the release of the parasitoid Diachsmimorpha longicaudata in different countries, against different  tephritid pests infesting host fruit species. This information is complementary to the new paragraph incorporated in the introduction.

New paragraph: “The establishment of D. longicaudata in the Americas has encouraged the augmentative biological control development not only against the exotic C. capitata [28-30], but also against economically important native Anastrepha species, such as Anastrepha suspensa (Loew) [31-32], A. ludens (Loew), A. obliqua (McQuart), A. serpentina (Wiedemann), A. striata (Schiner)  [23, 33-34], and A. fraterculus (Wiedemann) [35]”.

It is also important to note that the numbers equivalent to the references have been changed to restructure the position of the different references in the general list.

  1. Line 61, instead tephritid dipteran use taxonomic titles.

Response: “the tephritid dipteran” was removed from the text.

  1. Lines 94-98, add references

Response: references [19] was added.

  1. Lines 101-102 - add latin names of fruit species.

Response: species names were added to the following fruits included in the paragraph: grape (Vitis vinifera L.), olive (Olea europaea L.), stone fruits (apricot [Prunus armeniaca L. ], peach [Prunus persica (L.) Stokes] and plum [Prunus domestica L.]), fig (Ficus carica L.), quince (Cydonia oblonga Mill.).

  1. Line 121 - only reference 23 report use of D. longicaudata, references 26-28 not, please add correct reference.

Response: The reviewer's comment is partially correct. Both references [23] and [27] report the use of Diachasmimorpha longicaudta in release programs against tephritid pests. Original references [26] and [28] were changed by: “de Pedro, L.; Tormos, J.; Harbi, A.; Ferrara, F.; Sabater-Muñoz, B.; Asís, J.D.; Beitia, F. Combined use of the larvo-pupal parasitoids Diachasmimorpha longicaudata and Aganaspis daci for biological control of the medfly. Ann. Appl. Biol. 2019, 174, 40-50”, and “Ovruski, S.M.; Suárez, L.; Cancino, J.; Liburd, O.E.     Biological control of tephritid fruit flies in the Americas and Hawaii: a review of the use of parasitoids and predators. Insects 2020, 11, 662”, respectively.

  1. Materials and methods: Listing of the names and number of host plants per treated plots are confusing with local names, common names and latin names, please simplifie

Response: The corresponding paragraphs were simplified as requested by the reviewer.

  1. -2.1. - age of parasitiods releases.  

Response: The sentence “Female and male parasitoids released were 1-2 and 2-3 days old, respectively” was added in 2.1 subheading of Material and Methods section.

  1. Line 190 - what means little conventional chemical treatments? Explain.

Response: the sentence “Both farms have little or no conventional chemical treatment, rather aimed at commercial grape crops” was changed as follows: “Fruits were not exposed to phytosanitary treatments during parasitoid releases. Only herbicides were aimed for weed control in commercial grape crops”.

  1. Lines 198-201, state for which period is the FTD value reported, is it average for whole spring-autumn, since FTD is probably low in spring and high in autumn.

Response: the information requested by the reviewer is already provided in the paragraph: “Records on the number of caught medfly adults in the suburban areas of both TF and CF one year before the study revealed broad population variation of the pest between mid-spring and late autumn. Thus, the number of caught flies per trap and per day (= FTD index) ranged from 0.04 ± 0.04 to 0.89 ± 0.64 and from 0.09 ± 0.09 to 0.99 ± 0.30 in both TF and CT, respectively”.

  1. add the size of experimental plot in scheme 2 and 3.

Response: The reviewer was correct to information on the size of the experimental area, but we believe that the requested information should not be included in schemes 2 and 3. This is due to the fact that in scheme 4 the area of parasitoid release in the treatment farm is shown, and in schemes 7A-B [Treatment farm (A) and Control farm (B)] experimental areas with artificial devices with sentinel trap fruits is also shown. Therefore, it is appropriate 4 to incorporate the size of the release area in the heading of the Scheme 4 (“Scheme 4. Schematic representation of the parasitoid release transects in the Treatment farm covering a 6.4-ha release area”), as well as in the text […the farm, covering 6.4 ha (Scheme 4)]. Similarly, the sizes of the ''experimental areas'' were added under the heading of the scheme 7 [Scheme 7. Schematic representation of the transects with oviposition devices distribution in both Treatment farm (A) and Control farm (B), covering 5.8- and 6.1-ha experimental areas, respectively]. This information was included in the text “Devices covered around 58 and 61% of the total area of TF and CF, respectively (Scheme 7A-B)”.

  1. 2.3 section, explain better the release rate per release date/ per area size, not per bag. What is release rate females per ha? What is the interval of the release, is it 2 weeks?.

Response: First, the sentence: “Thus, between 528 and 600 parasitoids were released per bag” was changed by “Thus, each release bag held between 528 and 600 parasitoids”. Second, the release rate per release date/ per area size are indicated in the text, however the original sentence "Between ~16,000 and ~18,000 parasitoids (~1,692 ± 108 parasitoids/ha, mean ± SE) were released into TF per releasing period" has been changed as follows: “Between ~16,000 and ~18,000 parasitoids (~1,692 ± 108 parasitoids/ha including ~803.7 ± 126.9 females/ha, mean ± SE) were released in the TF per releasing time”. Third, although the text refers to the time between each release "Parasitoids were released every 12 to 13 days by ground…” it was decided to modify the sentence as follows: “The release interval was 12 to 13 days”.

  1. Add the size of area in Scheme 4.

Response: it was done.

Reviewer 2 Report

Comments on “Medfly population suppression through augmentative release of an introduced parasitoid in an irrigated multi-fruit orchard of central-western Argentina”

General comments

1  While the English is generally good, it needs polishing by a native English speaker as some parts do not read well.  For example, throughout the work, “releasing” is used where “release” would be the grammatically correct choice. (like “releasing dates” instead of “release date”).

Introduction

1  The Introduction is sound. One point not covered is the economics of augmentative using D. longicaudata.  What are the costs of its release, and what are the marketing advantages of this approach. Also, how is the cost of the parasitoids linked to the sterile male program, since that was the source of the rearing host.

Methods

1---Somewhere (Introduction of Methods) it needs to be clear if there are any wild hosts of Med fly in the study area in uncultivated plants, or is the only source the yard fruit in urban areas and various local crops.

2---Response data (fly trap catch and rates of parasitoid attack on pupae in sentinel fruits) were based on three and 10 sampling points, respectively in both the release and the control plots.  Why so few such points? That seems likely to not provide a good estimate of response parameters.

Results

1--- Why were no measures of infestation in the actual crops taken? That would seem to be the most easily comparable to other studies or other methods of control. It would also place the control level achieved in a general context that is easily understood. How close was the fruit in the release plot to being acceptable in either the national or international market?

Discussion

1--- The authors’ summary of the importance of their findings stressed two points “ two major findings stand out from this study. Firstly, the decrease in the medfly population at the D. longicaudata release site evidences the effectiveness of augmentative biological control using parasitoids on farms with little or no conventional chemical treatment. Secondly, parasitoids of the DlTSL-Cc lineage have a satisfactory performance once released under environmental conditions prevailing in fruit orchards, as they successfully found and parasitized medfly larvae”   I agree that both of these outcomes are supported by their data, but what is lacking is numerical context for each parameter: Was the reduction in Medflies, as measured by declines in trap catch, strong enough to protect the crop?  Was the cost of the biocontrol treatment per ha compared to costs of other control measures? 

Author Response

Dear Reviewer 2

Suggested corrections and changes by you have improved the quality of the manuscript. We thank  for their time in reviewing our manuscript. Below you will find a point-by-point response to the requests made. Best regards, Sergio M. Ovruski (Corresponding author)

Reviewer #2.

  1. While the English is generally good, it needs polishing by a native English speaker as some parts do not read well.  For example, throughout the work, “releasing” is used where “release” would be the grammatically correct choice. (like “releasing dates” instead of “release date”).

Response: a detailed revision of the text was carried out based on the reviewer's recommendation. Several sentence modifications, word changes, and grammatical restructuring were carried out

 The Introduction is sound. One point not covered is the economics of augmentative using D. longicaudata. What are the costs of its release, and what are the marketing advantages of this approach. Also, how is the cost of the parasitoids linked to the sterile male program, since that was the source of the rearing host.

 Response: the reviewer highlights an interesting and necessary issue for a biological control program within the context of integrated management of the target pest, namely the economic perspective. The San Juan Fruit Fly Control and Eradication Program is currently in the process of evaluating the cost-benefit of the techniques used for Medfly control in the province. The biological control cost-benefit analysis is not yet available. A comparative economic analysis with other control methods is also not yet available, but an approximate value of the parasitoid release bag is available. The cost per release bag is US$0.48, which may hold up to 600 parasitoids. This cost includes personnel salaries, rearing supplies, release equipment, and fuel for transporting parasitoids. It is also important to note that these scarce information on the cost of a parasitoid release bag is not available to the public (it has not been published yet), because it is confidential information belonging to the government of the province of San Juan; we are not authorized to provide it. Despite the foregoing, we do not consider that the manuscript should cover an economic aspect of the control method used. However, we believe it is important to point out the benefits of this method as indicated by the reviewer. For this reason we have included a paragraph in the introduction section highlighting this aspect of biological control, as follows:.

“In addition to this context, augmentative biological control as a well-used strategy is particularly desirable due to five key issues, as follows [25]: first, beneficial features, such as: environmentally and healthily safe, sustainable as it slows down the development of pest resistance, no phytotoxic damage to the crop, higher yields and a healthier product, reduction of pesticide residues; second, a developed biological control industry that facilitates large-scale and inexpensive mass production of biocontrol agents, with suitable quality control, effective packaging, distribution and release methods; third, biological control is effective in saving agricultural production when pesticides are not available because of environmental or human health concerns; fourth, requirements mainly from non-governmental organizations and consumers to reduce pesticide residues below the currently permitted thresholds; and fifth, global increase in policies aimed at the reduction and/or replacement of synthetic pesticides by more biological sustainable methods of pest management. Furthermore, a highly significant advantage of biological control is its compatibility with environmental and health standards, which allows growers to link with integrated pest management (IPM) and organic certification schemes [26]”.

Two new references were added to the text:

-van Lenteren, J.C.; Bolckmans, K.; Köhl, J.; Ravensberg, W.J.; Urbaneja, A. Biological control using invertebrates and microorganisms: Plenty of new opportunities. BioControl, 2018, 63, 39-59.

CABI Bioprotection portal. 5 advantages of biocontrol compared to chemical pest control. Available online: https://bioprotectionportal.com/es/blog/2022/5-advantages-of-biocontrol-over-chemical-control (accessed on 4 March 2023).

  1. Methods

3.a. Somewhere (Introduction of Methods) it needs to be clear if there are any wild hosts of Med fly in the study area in uncultivated plants, or is the only source the yard fruit in urban areas and various local crops.

Response: there were no wild hosts in the study area. We added a new sentence in subsection 2.2 of Materials and Methods (2.2. Experimental Locations and Study Area Description) as follows: “Backyard fruit and various local orchards were the only source of medfly in the study area”.

3.b. Response data (fly trap catch and rates of parasitoid attack on pupae in sentinel fruits) were based on three and 10 sampling points, respectively in both the release and the control plots.  Why so few such points? That seems likely to not provide a good estimate of response parameters.

Response: The choice of the number of release points we chose was based on the operational overhead involved plus insufficient technical support staff. However, in our opinion the choice of ten release points per transect was not a small one. We rely on this for the following reasons: (1) the three transects with the 10 release points each covered an area of 6.4 ha over a total of 10 ha, which is the size of the treatment farm, and which accounts for 64% of the total farm (this information was added to the text); (2) we focused on releasing in areas where hosts were available; and (3) results of releases showed a decrease in the medfly population, so the parasitoid was effective in finding and parasitizing host larvae.

  1. Results

Why were no measures of infestation in the actual crops taken? That would seem to be the most easily comparable to other studies or other methods of control. It would also place the control level achieved in a general context that is easily understood. How close was the fruit in the release plot to being acceptable in either the national or international market?

Response: the reviewer's point is very valid and we agree with the need to carry out a fruit survey to evaluate natural infestation levels at a comparative level. We fully share his opinion. However, there are two major drawbacks to developing such a method of pest control evaluation. First, we found the refusal of the people responsible from the respective study farm to remove the fruits from the site; this is mainly because the fruit, even the fallen ones, is normally used for the production of jams or jellies, and even as food for farm animals (pigs for example). Second, in a first evaluation we made with a very small fruit sample size (as allowed by the farm owners) did not give us promising results. Therefore, we decided to use artificial devices to hold a sentinel fruit and expose it to natural infestation, as described in the manuscript. This parasitoid release evaluation method was previously used on a commercial fig farm (where we could not remove fruit) with success. Regarding the query -How close was the fruit in the release plot to being acceptable in either the national or international market?-, we do not have this information, but it is important to take this into account in future parasitoid release studies.

  1. Discussion

The authors’ summary of the importance of their findings stressed two points “ two major findings stand out from this study. Firstly, the decrease in the medfly population at the D. longicaudata release site evidences the effectiveness of augmentative biological control using parasitoids on farms with little or no conventional chemical treatment. Secondly, parasitoids of the DlTSL-Cc lineage have a satisfactory performance once released under environmental conditions prevailing in fruit orchards, as they successfully found and parasitized medfly larvae”   I agree that both of these outcomes are supported by their data, but what is lacking is numerical context for each parameter: Was the reduction in Medflies, as measured by declines in trap catch, strong enough to protect the crop?  Was the cost of the biocontrol treatment per ha compared to costs of other control measures?.

Response: the two main findings meet the goal of this study based on the working hypothesis. For this study we had not performed an economic analysis of the impact of parasitoid releases as a crop protection factor, and we had not carried out a comparative cost-benefit analysis with other control methods. However, we believe that both approaches are important to evaluate and numerically demonstrate the effect of parasitoid releases. Unfortunately, we do not have this type of information. However, these economic analyses will be considered in the ongoing studies that are being carried out in the province of San Juan regarding SIT and Augmentative Biological Control against medfly.